# Comparative Analysis of Passing, Possession, and Goal-Scoring Trends in Euro 2024 and Copa America 2024

**DOI:** 10.3390/sports13100357

**Published:** 2025-10-09

**Authors:** Sattar Taheri-Araghi, Moji Ghadimi, Juan Del Coso

**Affiliations:** 1Department of Physics and Astronomy, California State University, Northridge, CA 91311, USA; 2Queensland Cyber Infrastructure Foundation, Brisbane, QLD 4072, Australia; moji131@gmail.com; 3Sport Sciences Research Centre, Rey Juan Carlos University, 28943 Fuenlabrada, Spain; juan.delcoso@urjc.es

**Keywords:** football analytics, match statistics, passing sequences, ball possession, goal-scoring patterns, tactical analysis

## Abstract

Football, as a team sport, relies on a delicate balance where tactical cohesion and strategic play are as critical as physical prowess. While evidence suggests that European teams often display higher physical intensity, the tactical differences between European and American football are still not well quantified. The aim of this study is to conduct a comparative analysis of passing, possession, and goal-scoring dynamics in Euro 2024 and Copa America 2024. Data from 51 Euro matches and 32 Copa America matches, encompassing all game events with sub-second precision, were obtained from StatsBomb. Analyses were performed in MATLAB, with possession calculated as ‘pure possession,’ excluding inactive periods. Euro 2024 teams demonstrated significantly more total passes per match (p<0.05, Cohen’s d=1.43), higher pass completion rates (p<0.05, Cohen’s d=1.30), and longer possession sequences (p<0.05, Cohen’s d=0.24). They also showed greater possession in the five minutes prior to goals (p<0.05, Cohen’s d=0.63). In contrast, Copa America 2024 teams favored longer passes (p<0.05, Cohen’s d=0.15), reflecting a more direct playing style. Possession disparities between teams in individual matches did not differ significantly (p=0.31, Cohen’s d=0.23), and the distribution of shot distances for goals was also similar across tournaments (p=0.86, Cohen’s d=0.02). In summary, Euro 2024 teams emphasized control through longer possession and greater passing accuracy, while Copa America 2024 teams relied on more dynamic and direct play. These findings underscore how regional footballing philosophies shape match strategies and outcomes, offering insights into the tactical diversity of international football.

## 1. Introduction

Football (soccer), the world’s most popular sport, captivates billions of spectators with its fluidity, unpredictability, and strategic depth. In recent decades, football analytics has evolved from anecdotal observations to a sophisticated, data-driven discipline. Early analyses focused on basic metrics such as goals, shots, and possession, but recent advancements in technology and data collection have enabled a deeper exploration of the game’s tactical and strategic elements [1,2]. Metrics such as passing sequences, possession dynamics, and goal-scoring trends are now recognized as fundamental to understanding team performance and strategic identity [3,4,5].

This has led to a new understanding of modern football, where match performance results from the complex dynamic interactions of physical, technical, and tactical actions and movements of all competing players [6]. Notably, some technical indicators have been found to predict team success more accurately than physical indicators [7], confirming the importance of assessing the variables related to the interconnectedness of players during gameplay [8]. In this regard, factors such as the number of shots, shots on target, number of passes, and pass completion rates have all been associated with team success [9,10]. Football performance is constantly evolving, with an upward trend in the frequency of high-intensity running and improvements in technical accuracy [11,12]. These physical and technical advancements are likely driven by various interrelated factors, including improvements in player physical preparation, emerging trends in tactical periodization, and the integration of modern technologies for competition monitoring and analysis [13]. Furthermore, analyzing how these technical indicators vary according to the type or context of competitions may provide valuable insights to help coaches better tailor training and tactical strategies to the specific demands of each competitive scenario. This approach could also assist in optimizing player selection and preparation based on the technical profiles most suited to a given competition.

In the national leagues of Europe, more successful teams tend to start gameplay from a more advanced position, engage in longer offensive sequences, progress at a slower speed, maintain greater ball possession, and exhibit higher passing accuracy [14,15]. High-ranking teams also implement a more combinative playing style, characterized by greater ball possession [16,17], increased game initiative [18], and reduced reliance on direct playing styles [19], compared to lower-ranked teams. Additionally, the most successful teams in national leagues complete a higher number of successful passes, while lower-ranked teams perform more defensive than offensive actions [14]. Several studies have defined and evaluated the effectiveness of multiple attacking strategies, such as positional attacks, fast attacks, direct attacks, and counterattacks [20], as well as different defensive strategies [21]. However, it remains unclear whether this playing style pattern is distinctive to European teams or whether it can be generalized to teams from other continents.

Metrics evaluating player interplay have also revealed distinct regional styles of play. European football, particularly in national leagues such as the Premier League and La Liga, emphasizes possession-based strategies characterized by structured build-up play and prolonged possession [22]. In contrast, American football often showcases a more dynamic, instinct-driven style, marked by rapid transitions, flair, and creativity [23]. These stylistic differences are deeply rooted in the cultural and tactical histories of their respective regions, influencing match outcomes and broader tournament trends [24]. Despite these established distinctions, comparative analyses of European and American football styles remain limited. Previous research has often focused on physical metrics, such as total distance covered [25], while tactical aspects like passing strategies and possession dynamics have received less attention.

The aim of this study is to conduct a comparative analysis of passing, possession, and goal-scoring dynamics in Euro 2024 and Copa America 2024. Since no previous study has comprehensively explored the tactical differences between European and American football teams, this investigation was conducted without a predefined hypothesis. Specifically, we investigated the following research questions: (i) Do European and American teams differ in their possession dynamics, including sequence length and pre-goal possession during national teams tournaments? (ii) Are there systematic differences in passing behavior, such as pass frequency, completion rate, and pass length distributions, between the two tournaments? (iii) Do goal-scoring patterns vary across regions? Addressing these questions provides a clearer understanding of how regional footballing philosophies shape match strategies and outcomes. By addressing these questions, the study aims to provide a clearer understanding of how regional footballing philosophies are reflected in measurable match outcomes.

## 2. Materials and Methods

### 2.1. Data Source and Acknowledgment

The data for this study were obtained from StatsBomb (Bath, UK), a leading provider of high-quality football data for research and analysis, which has made its data freely accessible to promote an analytical understanding of the game. We acknowledge that our use of StatsBomb data complies with their public data user agreement, and our conclusions remain independent of their opinions or insights. StatBomb’s data have been previously used in sports science literature for tactical analysis [26,27] and the accuracy of the data has been validated [28]. To conduct a comprehensive comparison of possession sequences, passing patterns, and goal-scoring dynamics, we analyzed detailed event data from Euro 2024 (24 teams, 51 matches) and Copa America 2024 (16 teams, 32 matches). This high-resolution dataset captures all match events with sub-second precision, allowing for rigorous quantitative comparisons between European and American teams with a standardized approach for both competitions. Using custom-developed MATLAB software (version R2022a, Natick, MA, USA), we calculated possession metrics, analyzed passing sequences, and quantified goal-scoring trends. The StatBomb data we used in this study were uploaded in July 2024 as open-data on Github.

### 2.2. Computed Parameters and Calculation Methods

This study involved the computation of key parameters to analyze possession, passing dynamics, and goal-scoring trends across every match in Euro 2024 and Copa America 2024. The overall workflow is summarized in Figure 1. Beginning with the raw StatsBomb JSON (JavaScript Object Notation) event data, match files were imported into MATLAB and harmonized to ensure consistent identifiers, event ordering, and coordinate systems. From these standardized data, three main analytical streams were developed: (i) possession timelines, capturing pure-possession percentages and their evolution across each game; (ii) passing features, including pass counts, completion rates, sequence lengths, and pass-length distributions; and (iii) goal analysis, identifying scoring events and their timing distributions. These analyses were then combined to evaluate pre-goal contexts, linking possession and passing sequences directly to goals, and finally aggregated at the tournament level to enable statistical comparisons between Euro 2024 and Copa America 2024. The following subsections describe the key parameters derived from the dataset and the methods used to compute them.

#### 2.2.1. Possession Parameters

**Possession Percentage:** Possession calculations in this study were based on *pure possession*—a metric that excludes dead-times such as stoppages, fouls, and substitutions when the ball is not in active play. Unlike traditional possession metrics, which often include the full duration of the match regardless of ball activity, this approach filters out inactive intervals to yield a more accurate representation of team control during active gameplay.

**Possession Disparity:** Possession disparity was calculated as the absolute difference between the final possession percentages of the two teams in each match. To assess trends, matches were ranked based on possession disparity, with the smallest differences placed at the top. This metric allowed for a direct comparison of how evenly possession was distributed in each tournament.

**Possession Segment Durations (P(t)):** The parameter P(t) was used to quantify the percentage of possession sequences that lasted longer than a given time threshold *t*, where *t* was measured in seconds. Possession segment durations were extracted from event data, and the proportion of sequences exceeding each threshold was plotted to compare the possession stability across tournaments.

**Possession Leading to Goals:** Possession data from the match timelines were analyzed to determine the average possession percentage of the goal-scoring team in the 5 min preceding each goal. The possession percentage was computed in progressively shorter time windows (e.g., last minute, last 2 min, etc.), and the values were averaged across all goals to evaluate the trends in possession immediately before scoring.

**Correlation between possession and goals:** The relationship between final possession percentage and the number of goals scored was examined using scatter plots and overlaid box plots. A statistical correlation test was performed using the pairwise comparisons of means to determine whether possession had a significant impact on goal-scoring outcomes.

#### 2.2.2. Passing Parameters

**Total Passes Per Game:** The total number of passes per game was computed by summing the number of passes attempted by both teams in each match. This metric was visualized using box plots to compare passing frequency across tournaments.

**Pass Completion Rate:** The pass completion rate was calculated as the ratio of successfully completed passes to total attempted passes for each match. A probability distribution function (PDF) was generated to illustrate the variation in pass success rates across different matches and tournaments.

**Pass Length Distribution:** Pass lengths were grouped into 2 m bins, and a probability distribution function (PDF) was created by normalizing the frequency of passes within each bin. This allowed for a statistical comparison of short- and long-pass preferences between tournaments.

**Passing Sequences:** Passing sequences were defined as the number of consecutive passes made by a team before losing possession. Each passing sequence was counted, and a probability distribution function (PDF) was constructed to compare passing sequence lengths. The analysis also categorized sequences into short (fewer than 5 passes) and long (5 or more passes) to highlight the differences in tactical styles.

**Pass Success Rate Over Time:** Pass success rates were analyzed separately for the first and second halves of each match. A temporal trend was plotted to assess how pass success rates evolved throughout each half period of the game, revealing variations due to fatigue or tactical shifts.

**Pass Success as a Function of Pass Length:** Passes were categorized based on their length, and the success rate for each category was computed. A probability curve was generated to illustrate how pass success rates varied across different pass distances, capturing trends in short-, medium-, and long-range passing effectiveness.

#### 2.2.3. Goal-Scoring Parameters

**Goals by Period:** The number of goals scored in each half and in extra time was recorded separately for both tournaments. This allowed for an assessment of whether teams were more likely to score early, late, or in extended play.

**Goals by Body Part:** The dataset recorded the body part used to score each goal (right foot, left foot, or head). A comparison was made between tournaments to evaluate differences in finishing styles.

**Shot Locations and Distance to Goal on the Pitch:** The coordinates of goal-scoring shots were extracted and mapped onto a scaled football field to visualize scoring trends. The Euclidean distance from the shot location to the center of the goal line was calculated for each goal. A probability distribution function (PDF) was generated to compare the typical goal-scoring distances between tournaments.

**Box Plot of Goal-Scoring Shot Distances:** Shot distances were visualized using box plots, displaying the median, interquartile range, and standard deviation of goal-scoring shot distances for both tournaments.

#### 2.2.4. Comparison of the Champions: Spain vs. Argentina

To explore stylistic and tactical differences between the tournament champions, we conducted a focused comparison of Spain (Euro 2024 winner) and Argentina (Copa America 2024 winner). For each team, game-level data were extracted to evaluate variation in possession percentage and total/completed passes across matches. Additionally, pass length distributions were used to produce probability distribution functions (PDFs), enabling a direct comparison of each team’s preferred passing range. Finally, pass success rates were used to assess the execution efficiency across distances. All metrics were derived using the same event-level dataset and processing pipeline as applied in the tournament analysis, ensuring consistency and comparability.

### 2.3. Statistical Analysis Approach

A rigorous statistical framework was employed to analyze possession, passing, and goal-scoring metrics, ensuring an accurate and insightful comparison between Euro 2024 and Copa America 2024.

To validate the accuracy of our custom MATLAB scripts, we performed several cross-checks. For a subset of matches (*n* = 7), possession timelines were plotted and manually compared with video footage, confirming that automated possession changes matched observed sequences of control. Goal timestamps were also verified against broadcast footage, with complete agreement in timing. Consistency checks ensured that team possession percentages summed to 100% and that total passes, shots, and goals matched official match summaries. Together, these steps provide assurance that the computed metrics reliably capture on-field events.

The core statistical method in this study was the pairwise comparison of estimated means and distributions, which provided a direct assessment of the differences between the two tournaments. Pairwise comparisons were performed using Welch’s two-sample *t*-tests and Kolmogorov–Smirnov tests in MATLAB, with significance reported at p<0.05. Welch’s *t*-test was chosen because it does not assume equal variances between groups, and its use was complemented by nonparametric tests (Kolmogorov–Smirnov) to account for deviations from the normality observed in several variables. The independence of observations was ensured by treating each match (or event) as a separate unit. Statistical significance threshold was set at p<0.05. In practice, all *p*-values obtained were either below 0.001 or well above 0.05, so the choice of threshold does not alter the conclusions. Effect sizes (quantified using Cohen’s *d* for standardized mean differences) were also provided to reflect the magnitude of observed differences. In reporting effect sizes, we interpreted Cohen’s *d* by magnitude using negligible (|d|<0.10), small (0.10≤|d|<0.30), moderate (0.30≤|d|<0.50), moderate-to-large (0.50≤|d|<0.80), large (0.80≤|d|<1.20), and very large (|d|≥1.20); this follows standard practice based on Cohen’s conventional 0.2/0.5/0.8 benchmarks with commonly used extensions. This combined approach allowed for a robust assessment of the differences in possession, passing, and goal-scoring trends while avoiding overreliance on strict parametric assumptions.

In addition, Kendall’s Tau correlation analysis was conducted to assess the strength and direction of association between selected ranked variables, providing a non-parametric measure of correlation suitable for ordinal and non-normally distributed data.

To support these analyses, probability distribution functions (PDFs) were used to examine the distribution of key parameters, such as pass lengths and goal-scoring shot distances. Each metric was binned into appropriate intervals (e.g., 2 m bins for pass lengths), and the frequency of observations was normalized to generate probability distributions. These PDFs facilitated direct comparisons of passing and shooting behaviors across the two tournaments, offering insights into tactical differences.

### 2.4. Ethical Considerations

All analyses adhered to ethical guidelines for the use of publicly available sports data, as outlined in the StatsBomb public data user agreement. The data were utilized solely for non-commercial, research purposes, and all conclusions were derived independently by the authors.

## 3. Results

### 3.1. Possession Statistics

#### 3.1.1. Possession Disparity

Figure 1B and Figure 2A show possession disparity, defined as the difference between the final possession percentages of the two teams in each match. In Figure 2A, matches are ranked by the extent of possession disparity, with the smallest disparity at the top. Data points are color-coded, with Copa America 2024 represented in light blue and Euro 2024 in dark orange. Teams with higher possession are plotted to the right of the 50% line, while teams with lower possession are plotted to the left; as such, two data points are indeed collected from each match. 

Figure 2B presents the distribution of possession percentages in both tournaments using horizontal box plots. The interquartile range, median, and spread of possession percentages are shown for each competition. The analysis is based on “pure possession,” which excludes dead time, as defined in the Methods section. While the average value of possession is expected to be 50%, the distribution around the average can point out differences between the two competitions. However, the data from Figure 2A,B indicate no considerable difference in the distribution of possession percentages between Copa America 2024 and Euro 2024. A Kolmogorov–Smirnov (K.–S.) test comparing possession disparity between the two tournaments resulted in a p=0.31, confirming the absence of a statistically significant difference. Consistently, the effect size was small (Cohen’s d=0.23), indicating only a modest practical difference. Full descriptive statistics and test outcomes are reported in Table 1.

#### 3.1.2. Correlation Between Possession and Goals

Figure 2C examines the correlation between the final possession percentage and the number of goals scored by each team. Scatter plots and overlaid box plots illustrate this relationship, with the Y axis representing the number of goals scored (0–5) and the X axis showing the corresponding final possession percentage. The box plots display the distribution of possession percentages among teams that scored the same number of goals, revealing a wide range of possession values across different goal counts.

A Kendall’s Tau correlation analysis was conducted to assess the relationship between the possession percentage and goals scored in Copa America 2024 and Euro 2024. In Copa America, the correlation was weak and not statistically significant (τ = 0.166, *p* = 0.083), indicating no meaningful association between possession and goal-scoring. In Euro 2024, the correlation remained weak (τ = 0.155) with negligible statistical significance (*p* = 0.040), suggesting a slight tendency for teams with higher possession to score more goals. However, the low correlation values in both tournaments indicate that the possession percentage alone is not a strong predictor of goal-scoring.

#### 3.1.3. Possession Segment Durations and Possession Leading to Goals

Figure 2D shows the distribution of possession sequence durations for Copa America 2024 and Euro 2024. The parameter P(t) denotes the percentage of possession sequences that lasted longer than a given time threshold *t*, measured in seconds. This figure illustrates how the possession stability differs between the two tournaments across various duration thresholds.

The orange curve (representing Euro 2024) and the blue curve (representing Copa America 2024) allow for a direct visual comparison of P(t) across all time values. Notably, the P(t) values for Euro 2024 remain consistently higher than those for Copa America 2024 at any given *t*, suggesting that Euro teams sustained longer possession sequences. This method of comparing entire curves provides an intuitive, distribution-level assessment of possession stability between the tournaments. The statistical comparison in Table 1 also shows that Euro 2024 teams sustained longer possessions on average (17.6 ± 19.5 s) compared with Copa America 2024 (14.3 ± 16.2 s), with a significant statistical difference (K.–S. test, p<0.05). However, the effect size was small (Cohen’s d=0.18), indicating that the practical magnitude of the difference is modest.

Figure 2E illustrates the average possession percentage of the goal-scoring team during the 5 min period leading up to each goal. The X axis represents time relative to the goal, ranging from 5 min before the goal (t=−5) to the moment the goal is scored (t=0), while the Y axis shows the average possession percentage of the scoring team within each time window. The data points forming the solid curves in the figure correspond to progressively shorter time windows (e.g., last minute, last 2 min, etc.), with possession percentages averaged across all goals for each tournament.

The curves indicate that, at every time interval analyzed within the 5 min window, Euro 2024 teams consistently held a higher possession percentage than Copa America 2024 teams prior to scoring, suggesting a more sustained control of the ball in the lead-up to goals. Statistical analysis (Table 1) using Welch’s two-sample *t*-test—chosen instead of the Kolmogorov–Smirnov test because the comparison involved mean possession values from two independent groups rather than full distributional shapes—confirmed that the difference was highly significant (p<0.05) with a moderate-to-large effect size (Cohen’s d=−0.63), indicating that Euro teams systematically maintained higher possession prior to goals than their Copa counterparts.

### 3.2. Passing Dynamics

#### 3.2.1. Total Passes and Pass Completion Rate per Game

Figure 3A compares the total number of passes per game in Copa America 2024 and Euro 2024. The box plots illustrate the distribution of passes per game by summing the number of passes attempted by both teams in each match. Euro 2024 matches featured substantially more passes per game than Copa America 2024, with the difference confirmed by statistical analysis (Table 2). The K.–S. test indicated a highly significant difference (p<0.05), and the effect size was very large (Cohen’s d=1.43), showing that Euro 2024 matches consistently involved far greater passing activity compared with Copa America 2024 matches.

Figure 3B presents the distribution of pass completion rates across both tournaments using probability distribution functions (PDFs). Euro 2024 exhibited consistently higher completion rates, with the distribution concentrated around higher percentages compared to Copa America 2024, which showed a broader spread and lower central tendency. Statistical comparison using the K.–S. test confirmed a highly significant difference between the two distributions (p<0.05). As summarized in Table 2, the effect size was very large (Cohen’s d=1.30), indicating that Euro matches systematically achieved a higher passing accuracy than Copa America matches.

#### 3.2.2. Pass Length Distribution

Figure 3C shows the probability distribution of pass lengths in Copa America 2024 and Euro 2024. Pass lengths were grouped into 2 m bins, and the resulting data were normalized to generate a probability density function (PDF). Overall, Euro 2024 exhibits a greater concentration of shorter passes, while Copa America 2024 displays a broader and more balanced distribution across different pass lengths.

The inset of Figure 3C further highlights this contrast by dividing passes into two categories: short passes (<20 m) and long passes (>20 m). Euro 2024 teams show a clear preference for short passes, while Copa America 2024 teams tend to distribute passes more evenly between short and long distances.

Statistical comparison of pass lengths between the two tournaments confirmed these visual trends. As reported in Table 2, Copa America passes were on average longer (22.3 ± 15.5 m) than those in Euro 2024 (20.1 ± 13.6 m), and the Kolmogorov–Smirnov test indicated a highly significant difference between the distributions (p<0.05). However, the effect size was small (Cohen’s d=0.15), suggesting that, while Copa teams systematically favored longer passes, the practical magnitude of the difference was modest.

#### 3.2.3. Passing Sequences

Figure 3D shows the probability distribution of the number of passes per sequence. The X axis represents the number of passes in a sequence, while the Y axis indicates the likelihood of observing a sequence of that length. In both tournaments, shorter passing sequences—where possession ends after just a few passes—are more common than longer ones.

The median number of passes per sequence was 3 (IQR = 5) for Copa America 2024 and 4 (IQR = 6) for Euro 2024. While both tournaments exhibited wide distributions, the Kolmogorov–Smirnov test confirmed a statistically significant difference between them (p<0.05). As shown in Table 2, the effect size was small (Cohen’s d=−0.24), indicating that, although Euro teams systematically sustained longer passing sequences than Copa teams, the magnitude of this difference was modest in practical terms.

Comparatively, Copa America 2024 displays a higher probability of shorter passing sequences, while Euro 2024 teams tend to sustain possession for longer sequences. The inset of Figure 3D highlights this contrast by separating passing sequences into short (fewer than five passes) and long (five or more passes), further illustrating the difference in distribution between the two tournaments.

#### 3.2.4. Variation in Pass Success by Match Time and Pass Length

Figure 3E shows how pass success rates evolve over the course of a match, with separate analyses for the first and second halves. Euro 2024 teams maintain higher pass success rates across both halves. In both tournaments, pass success rates increase at the beginning of each half and decline in the final minutes. The lowest pass success rates are observed in the last 5–10 min of the second half in both competitions. The extra-time data are not included in this part of the analysis because the sample size was small with Copa America 2024 only having one match with extra-time.

Figure 3F presents the relationship between pass length and pass success rates. The success rate is lower for very short passes (less than 5 m), increases for passes in the 5–20 m range, and gradually declines for longer passes. Both tournaments exhibit this pattern.

### 3.3. Goal Scoring and Shot Distribution

#### Goal-Scoring Patterns: Timing, Body Part, and Shot Location on the Pitch

Figure 4A shows the distribution of goals scored in the first half, second half, and extra time for both Copa America 2024 and Euro 2024. In both tournaments, more goals were scored in the second half than in the first half. Extra time accounts for a small portion of the total goals (note that Copa America 2024 did not have extra-time regulation except for the final match, in which one goal was scored).

Figure 4B presents the number of goals scored by body part (right foot, left foot, and head) for both tournaments. The right foot accounts for the highest number of goals, followed by the left foot and then headers, a trend that is observed in both competitions.

Figure 4C shows the spatial distribution of goal-scoring shots (excluding own goals but including penalty shots) for both tournaments, with each shot location marked as a circle on a scaled football field.

Figure 4D compares the probability distribution of shot distances, defined as the Euclidean distance from the shot location to the center of the goal line, for Copa America 2024 and Euro 2024. A distinct peak is observed around the penalty spot, corresponding to goals scored from penalty kicks during regular play (penalty shootouts excluded). Statistical analysis (Table 3) confirmed that the two distributions were statistically indistinguishable, with the Kolmogorov–Smirnov test yielding p=0.64. The effect size measures were negligible (Cohen’s d=0.03), indicating that shot distance patterns resulting in goals were essentially identical across the two tournaments.

Figure 4E presents a box plot comparing shot distances between the two tournaments. The plots display the median, interquartile range, mean, and standard deviation, highlighting the similarity in both the central tendency and variability of goal-scoring shot distances across Copa America 2024 and Euro 2024.

### 3.4. Comparison of the Champions: Spain vs. Argentina

Possession Dynamics: Figure 5A provides a game-by-game analysis of possession percentages for Argentina in Copa America 2024 and Spain in Euro 2024. Possession percentages fluctuate from match to match for both teams. Spain’s possession against Croatia was below 50%, corresponding to an early lead in that match. Statistical comparison confirmed no significant difference between the two teams’ possession percentages (K.–S. test, p=0.69, Cohen’s d=0.23), indicating broadly similar possession profiles (Table 4).

Passing Patterns: Figure 5B presents the number of passes—both total and completed—executed by Argentina and Spain throughout their respective tournaments. The number of passes varies from game to game. Spain’s total pass count against Croatia was lower despite a decisive victory. The statistical analysis showed no significant difference in total passes per game for Spain vs. Argentina (K.–S. test, p=0.33, Cohen’s d=0.56).

Pass Length Distribution: Figure 5C presents the probability distribution function (PDF) of pass lengths for Argentina and Spain. The analysis shows that both teams have a similar distribution of pass lengths, with overlapping PDF curves. Nevertheless, statistical comparison indicated a significant difference in pass lengths (Kolmogorov–Smirnov test, p<0.05, Cohen’s d=0.08), although the effect size was negligible, confirming that practical differences were minimal (Table 4).

Pass Success Rate by Length: Figure 5D shows the pass success rate as a function of pass length for both teams. Argentina and Spain performed similarly in pass success rates across different distances. The success rate curves overlap, showing comparable effectiveness in executing passes. Consistently, statistical testing found no significant difference in overall pass success rates (Welch’s *t*-test, p=0.23, Cohen’s d=0.75), indicating broadly similar passing efficiency between the two champions (Table 4).

## 4. Discussion

This study presents a comparative analysis of possession dynamics, passing sequences, and goal-scoring patterns in Euro 2024 and Copa America 2024, providing insights into tactical differences between European and American international football. Overall, the findings indicate clear distinctions in playing styles, with Euro 2024 teams demonstrating more structured possession-based strategies, whereas Copa America 2024 teams emphasized direct play and rapid transitions. More specifically, this study revealed the following: (a) despite similar overall possession percentages across teams in each tournament, Euro 2024 teams exhibited a higher proportion of possession sequences across all time thresholds analyzed, suggesting a greater use of prolonged combinative play to reach the opponent’s goal; (b) Euro 2024 teams completed more passes per game, with longer sequences and higher passing accuracy, indicating a more refined and cohesive passing strategy compared to Copa America 2024 teams; (c) Euro 2024 teams relied more heavily on short passes—both those under 5 m and those under 20 m—and demonstrated higher accuracy across most pass length categories, suggesting superior passing efficiency and a tactical preference for short-passing combinations; (d) Euro 2024 teams also maintained higher possession in the minutes preceding goals, pointing to a more structured and deliberate build-up to scoring compared to the more direct style seen in Copa America 2024; (e) despite these tactical differences, there were no significant differences between tournaments in total goals per match, distribution of goals by half, scoring locations on the pitch, or the body parts used to score. These findings highlight the influence of regional footballing philosophies on tactical execution and underscore the importance of analyzing technical variables within the specific context of each competition. Specifically, the main outcomes of the study are consistent with the theoretical frameworks of game models that distinguish between combinative/positional play and direct/transition-based styles [16]. Our results support prior evidence from European domestic leagues, where successful teams emphasize structured build-up and possession control [14,15], while also aligning with studies showing that South American teams adopt more vertical and instinct-driven attacking approaches [23,24]. By situating our results within these established models, we reinforce the idea that tournament contexts reflect broader, culturally embedded tactical philosophies.

### 4.1. Possession and Passing Sequences: Structured Play vs. Direct Play

Possession analysis revealed that Euro 2024 teams maintained longer possession sequences compared to Copa America 2024 teams, reflecting a more controlled and sustained style of play. This may be a pattern not only distinctive of European national teams, as this type of more associative style of play that includes longer passing sequences has been described in national leagues such as the Spanish LaLiga [11]. In contrast, Copa America 2024 teams engaged in shorter, more dynamic possession sequences, consistent with a transition-based tactical approach. These findings align with previous studies suggesting that European teams prioritize positional play and systematic ball circulation, whereas American teams rely more on quick counterattacks and fluid attacking movements. This may be because European teams—even at academic and developmental levels [29]—emphasize possession control and long passing sequences as key performance indicators. As a result, the use of direct attacking strategies may become less effective in professional football, particularly in terms of creating goal-scoring opportunities [30]. American teams often demonstrate greater effectiveness in direct play, a style characterized by swift transitions and vertical attacks. This approach aligns with the tactical preferences observed in certain American national teams. Moreover, the physical demands associated with such a style are evident in the match performances of American teams. A study analyzing the 2019 Copa America 2024 highlighted the significant physical and technical demands placed on players, particularly in positions requiring rapid transitions and high-intensity efforts [31]. Despite these differences, possession percentage disparities between teams were comparable across both tournaments, indicating that the range of possession dominance across matches was similar. Furthermore, possession percentage was not significantly correlated with goal-scoring outcomes (or weakly in Euro 2024), reinforcing previous findings suggesting that high possession does not directly translate to scoring efficiency [32]. These results suggest that teams can be successful with varying degrees of possession, depending on their tactical execution and transition effectiveness. Collectively, this study highlights distinct tactical styles between Euro 2024 and Copa America 2024 teams, with European teams favoring longer, controlled possession sequences and American teams relying more on direct, transition-based play. Despite these differences, possession percentage showed little correlation with goal-scoring, suggesting that both possession-heavy and direct styles can be effective when well executed. These findings reinforce the value of tactical adaptability over a one-size-fits-all approach to match success [33]. Therefore, differences in pitch dimensions and surface quality may also have influenced our findings, as smaller or less consistent pitches can reduce passing accuracy, shorten possession sequences, and encourage longer passes, whereas larger, higher-quality pitches facilitate structured build-up play.

The passing analysis supports these distinctions in tactical approaches. Euro 2024 teams attempted a higher number of passes per game and demonstrated a preference for short, controlled passing sequences, reinforcing a structured build-up approach. In contrast, Copa America 2024 teams exhibited greater variability in passing length and sequence duration, indicating a greater reliance on mixed passing strategies and direct play. This may not be a new characteristic of these competitions, as one-touch finishing was shown to enhance goal-scoring opportunities compared to multi-touch attempts in Copa America 2024, suggesting a sustained presence of this direct style of play over time [34]. On the other hand, in Euro 2024, it has been found that winning teams had more assists, attempts on target, and runs into the penalty area. Furthermore, defensive metrics showed that winning teams in Euro 2024 recovered more balls [35]. The difference in pass completion rates further highlights these distinctions, with Euro 2024 teams exhibiting more consistent and precise passing accuracy across all distances, while Copa America 2024 teams showed greater variation in passing. These findings underscore how regional playing styles are reflected in passing behavior, with European teams favoring consistency and control, typically associated with a slower overall match tempo. In contrast, American teams exhibit greater tactical flexibility and spontaneity in possession, contributing to the perception of a faster pace and higher intensity. The causes of these differences are likely multifactorial and remain to be thoroughly investigated. While there are currently no scientific studies directly addressing this issue, media reports have highlighted potential contributing factors such as smaller pitch dimensions and lower pitch quality in some American venues compared to those used in Euro 2024 [36].

### 4.2. Goal-Scoring Patterns and Shot Distribution

The analysis of goal-scoring trends revealed that, in both tournaments, more goals were scored in the second half than in the first, a pattern consistently reported in elite-level football [37]. This trend may be attributed to increased attacking intent as the match progresses, tactical adjustments made during halftime, or cumulative defensive fatigue [38]. Goals scored in extra time represented only a small fraction of the total, reinforcing the critical nature of regular-time scoring efficiency. Additionally, differences in goal-scoring shot distances between Euro 2024 and Copa America 2024 were minimal, with both tournaments displaying similar patterns. Notably, a peak in shot distance distributions corresponding to penalty goals was observed in both cases, as expected [39]. These findings suggest that, while tactical styles and match tempo may differ across competitions, certain temporal and spatial goal-scoring trends remain consistent at the elite level.

### 4.3. Comparing the Champions: Spain vs. Argentina

A comparison between Spain (Euro 2024 winner) and Argentina (Copa America 2024 winner) highlights the similarities in passing and possession metrics, despite the differences in the competitive environments of the two tournaments. Both teams exhibited variability in possession across matches, suggesting that they adapted their ball control strategies based on the opponent and game context. Spain’s possession percentage against Croatia was notably lower than in other matches, a match where Spain took an early lead. This trend aligns with the notion that teams with an early advantage often adjust their tactical approach, allowing the opposition more possession while maintaining defensive stability [32]. Despite variations in match contexts, both Spain and Argentina displayed similar pass length distributions, with significant overlap in their probability distribution functions. This suggests that both teams favored a well-balanced mix of short, medium, and long passes, supporting the notion that elite-level teams tend to optimize passing structure rather than adhere to a single dominant passing length category. Furthermore, pass success rates for both teams were comparable across distances, as successful teams maintain a high level of passing efficiency regardless of their tournament environment.

### 4.4. Limitations

This study has several limitations that should be acknowledged. First, the analysis is based exclusively on data from two tournaments—Euro 2024 and Copa America 2024—which, while representative of elite-level international competition, may not fully capture the broader tactical and stylistic characteristics of European and American football as a whole. National team tournaments occur under unique conditions, with limited preparation time and squad rotation, which may not reflect the day-to-day tactical behaviors observed in domestic leagues [40]. Second, the findings are specific to professional men’s football and may not be generalizable to other categories such as youth, amateur, or women’s football, where physical, technical, and tactical dynamics can differ considerably. Third, there is a growing number of American players competing in European leagues [41]. This may lead to a degree of stylistic convergence, as these players are increasingly influenced by European tactical and training methodologies, potentially diluting regional distinctions. Additionally, the analysis did not consider contextual match variables such as opposition quality, scoreline status, or game state, all of which can significantly influence tactical choices [42]. It should be also acknowledged that tournament-specific factors such as match scheduling, climate conditions, travel requirements, or referee styles may have influenced the tactical behaviors observed, and future studies should explicitly analyze the extent to which these contextual elements shape performance outcomes. Finally, while the study identified differences in possession, passing, and goal-scoring trends, it relied on aggregate tournament data and did not include qualitative tactical analysis, which could offer a deeper understanding of decision making and in-game adaptations.

### 4.5. Practical Applications

From a practical perspective, the findings of this study offer valuable guidance for coaches aiming to align training methodologies with distinct tactical identities. For teams favoring a possession-based style, as observed in Euro 2024, training should emphasize structured build-up play, passing accuracy under pressure, and positional “rondos” to develop players’ decision making in tight spaces. Coaches may incorporate multi-phase passing drills that replicate the slower, more deliberate tempo of controlled possession, promoting patience and tactical discipline in maintaining ball circulation. Conversely, for teams employing a direct or transition-based style, like many Copa America 2024 teams, training should prioritize rapid ball progression, vertical passing, and high-intensity counterattack simulations. Exercises such as wave attacks or small-sided games with directional constraints can condition players to exploit space quickly and make decisive attacking actions. Importantly, coaches should also consider hybrid approaches, using match analysis to fine-tune training to the tactical demands of different opponents. As highlighted by Hughes and Franks (2005) [43], the nature of passing sequences and their relation to goal-scoring offers critical insight into the effectiveness of different playing styles. Tailoring training not just for a team’s identity, but to the contextual nuances of the game, can be a decisive factor in performance at the elite level.

### 4.6. Public-Health Relevance

Although this study focuses on tactics, the playing styles we observe and report (possession-based versus transition-based) imply different external loads on the players. Under congested schedules, higher loads are associated with increased match-injury incidence [44]. Summer tournament conditions also elevate heat strain; consensus guidance recommends acclimation, cooling breaks, and time-of-day adjustments to reduce risk [45].

## 5. Conclusions

This study provides a quantitative comparison of tactical approaches in Euro 2024 and Copa America 2024, highlighting distinct differences in possession control, passing sequences, and goal-scoring patterns. The results demonstrate that Euro 2024 teams exhibited a more structured, possession-based playing style, characterized by longer passing sequences and sustained ball retention, while Copa America 2024 teams relied more on rapid transitions and mixed passing strategies. This study contributes to the growing body of research on football performance analysis, offering insights into the interplay between possession, passing structure, and goal-scoring at the highest levels of international competition. Future research could further examine the impact of defensive structures on possession-based vs. transition-based teams, as well as the influence of match context (e.g., game state, opponent strength) on passing and goal-scoring behaviors. Still, football is not a sport characterized by fixed or stable behavior; rather, it evolves continuously like a living organism, adapting to new tactical, physical, and technical demands over time [46]. Therefore, the ongoing analysis of national team competitions is essential to identify and understand emerging trends and the evolution of the game at the international level.

## Figures and Tables

**Figure 1 sports-13-00357-f001:**
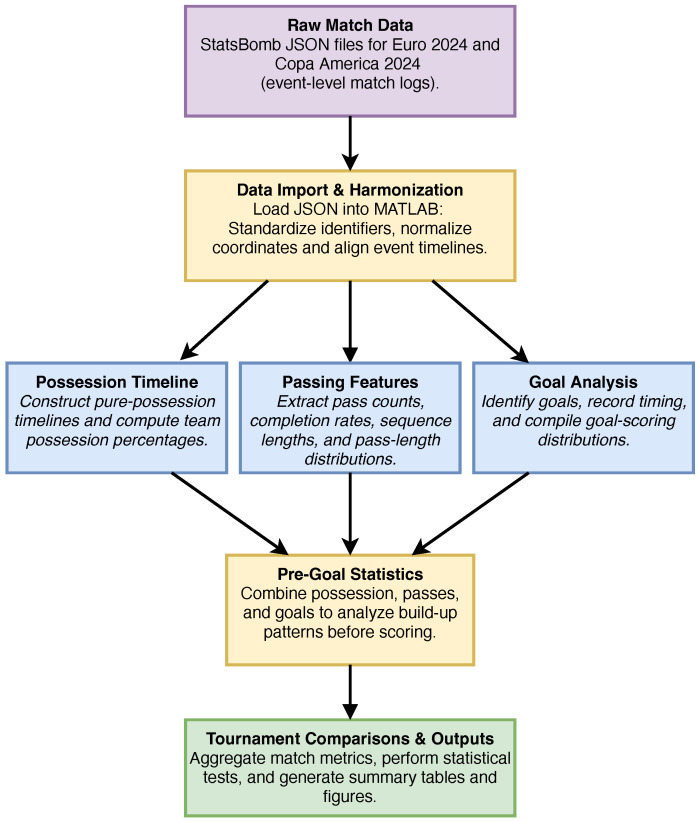
Analysis Pipeline for Euro 2024 and Copa America 2024. Flowchart showing the progression from raw StatsBomb match data through data import and harmonization, followed by analyses of possession timelines, passing features, and goal events. These analyses were integrated to assess pre-goal contexts and then aggregated to tournament-level outputs.

**Figure 2 sports-13-00357-f002:**
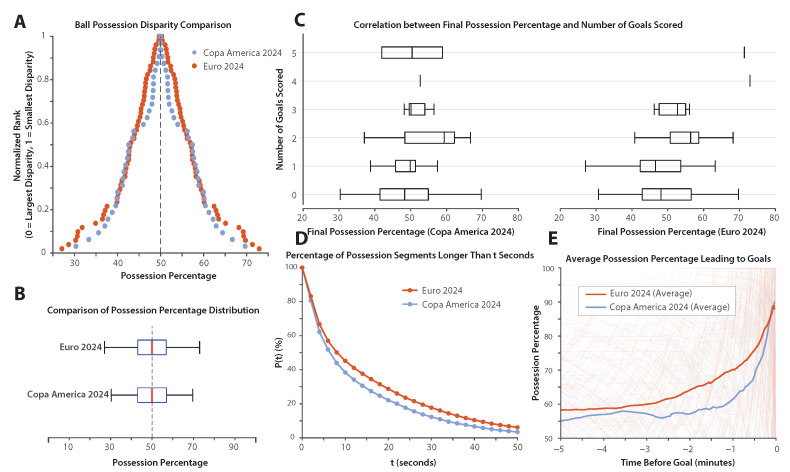
Comparative possession and goal-scoring analysis in Copa America 2024 and Euro 2024. (**A**,**B**) Possession disparity between teams in each match. Panel A ranks matches by possession disparity, with higher possession teams plotted to the right of 50%; as such, two datapoints are illustrated for each match. Panel B uses horizontal box plots to illustrate the distribution of possession percentages in both tournaments. (**C**) Correlation between goals scored and possession percentage. Scatter and box plots show the relationship between possession percentage and the number of goals scored. (**D**) Comparison of the possession segment durations. The parameter P(t) shows the percentage of possession sequences lasting longer than a specified time threshold *t*. (**E**) Possession percentage leading to goals. The average possession percentage in the 5 min preceding a goal is plotted for both tournaments.

**Figure 3 sports-13-00357-f003:**
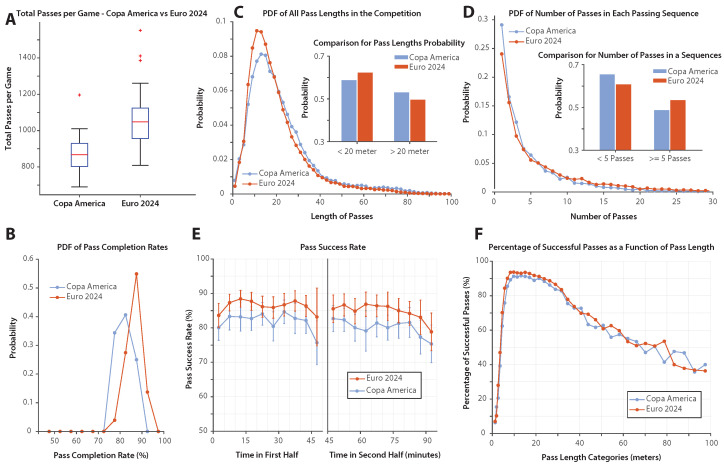
Passing dynamics in Copa America 2024 and Euro 2024. (**A**) Comparison of the total number of passes per game between Copa America 2024 (blue) and Euro 2024 (orange). Euro 2024 teams display a higher median number of passes per game. (**B**) Probability distribution function (PDF) of pass completion rates in both tournaments. Euro 2024 demonstrates a higher concentration of games with high pass success rates, while Copa America 2024 exhibits greater variability in pass accuracy. (**C**) The PDF of pass lengths in Copa America 2024 and Euro 2024. Euro 2024 teams have a higher probability of shorter passes, whereas Copa America 2024 shows a more balanced distribution across various pass lengths. Inset: A comparison of the probability of passes shorter than 20 m and those longer than 20 m. (**D**) PDF of the number of passes in each passing sequence. Euro 2024 has a higher probability of longer passing sequences, while Copa America 2024 has a higher probability of shorter passing sequences. Inset: A comparison of passing sequences with fewer than 6 passes and those with more than 6 passes. (**E**) Pass success rates over the course of matches, shown separately for the first and second halves. Euro 2024 teams maintain higher pass success rates throughout both halves, with a decline in the final minutes. (**F**) Percentage of successful passes as a function of pass length. Euro 2024 teams maintain higher pass success rates across all pass lengths.

**Figure 4 sports-13-00357-f004:**
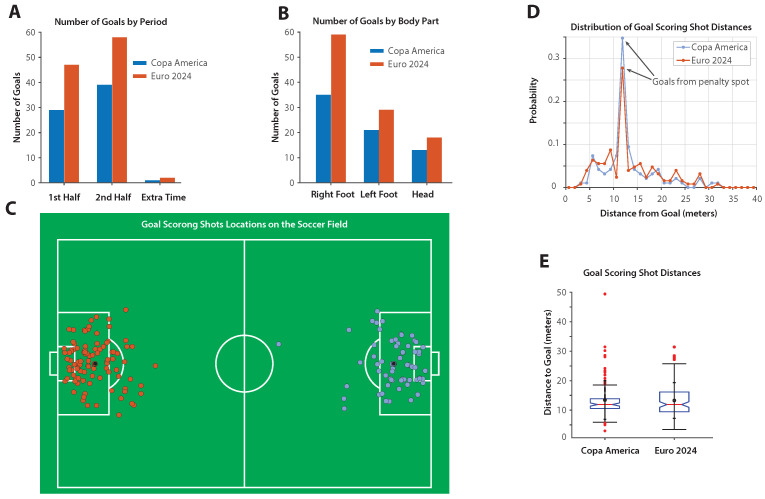
Goal scoring and shot distribution in Copa America 2024 and Euro 2024. (**A**) Goals by period: A comparison of goals scored in the first half, second half, and extra time in Copa America 2024 (blue) and Euro 2024 (orange). (**B**) Goals by body part: A comparison of the number of goals scored using the right foot, left foot, and head in both tournaments. (**C**) Shot locations on the football field: A comparison of the locations of goal-scoring shots, plotted on a scaled football field. (**D**) Distribution of goal-scoring shot distances: A probability distribution comparing the distances of goal-scoring shots. (**E**) Box plot of goal-scoring shot distances: A comparison of the distribution, median, and standard deviation of goal-scoring shot distances.

**Figure 5 sports-13-00357-f005:**
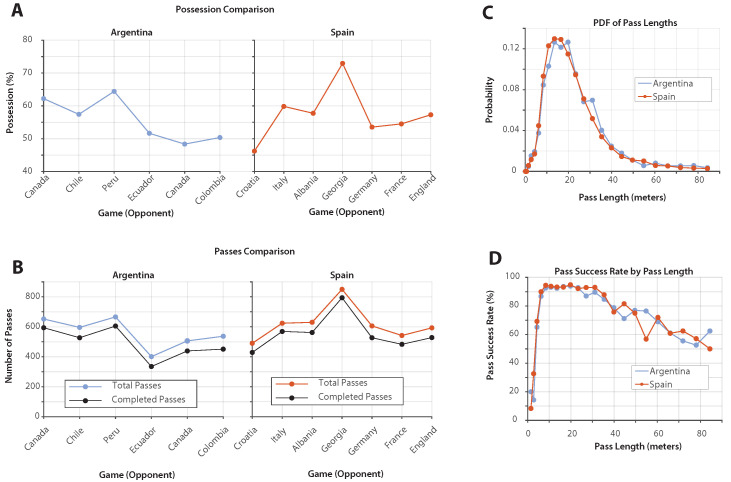
Comparison of Spain and Argentina’s performances in Euro 2024 and Copa America. (**A**) Possession by game: Comparison of Argentina’s possession percentages in Copa America 2024 (**left**) and Spain’s possession percentages in Euro 2024 (**right**), showing variability influenced by opponents and match dynamics. (**B**) Passes by game: Total and completed passes for Argentina (**left**) and Spain (**right**) across their tournaments, highlighting game-specific variability in passing patterns. (**C**) PDF of pass lengths: Distribution of pass lengths for Argentina and Spain, indicating similar passing profiles with overlapping distributions. (**D**) Pass success rate by length: Success rates of passes by length for both teams, showing comparable effectiveness in executing passes across different distances.

**Table 1 sports-13-00357-t001:** Possession statistics comparison between Copa America 2024 and Euro 2024.

Parameter	Copa (Mean ± SD)	Euro (Mean ± SD)	Test	*p*-Value	Effect Size (Cohen’s *d*)
Possession disparity per match (%)	13.0 ± 10.6	15.6 ± 11.6	K.–S. test	0.31	0.23
Possession segment duration (s)	14.3 ± 16.2	17.6 ± 19.5	K.–S. test	2.6 $\times \;10^{-28}$	0.18
Possession 5 min prior to goals (%)	60.1 ± 7.0	64.6 ± 7.4	Welch *t*-test	7.2 $\times \;10^{-14}$	0.63

**Table 2 sports-13-00357-t002:** Passing statistics comparison between Copa America 2024 and Euro 2024.

Parameter	Copa (Mean ± SD)	Euro (Mean ± SD)	*p*-Value (K.–S. Test)	Effect Size (Cohen’s *d*)
Total passes per game	871 ± 105	1057 ± 143	6.5 $\times \;10^{-7}$	1.43
Pass completion rate (%)	82.0 ± 3.5	86.2 ± 3.0	5.0 $\times \;10^{-5}$	1.30
Pass length (m)	22.3 ± 15.5	20.1 ± 13.6	6.8 $\times \;10^{-57}$	0.15
Passes per sequence	4.8 ± 5.4	6.4 ± 7.5	2.3 $\times \;10^{-27}$	0.24

**Table 3 sports-13-00357-t003:** Goal scoring statistics comparison between Copa America 2024 and Euro 2024.

Parameter	Copa (Mean ± SD)	Euro (Mean ± SD)	*p*-Value (K.–S. Test)	Effect Size (Cohen’s *d*)
Goal-shot distance (m)	13.4 ± 6.6	13.3 ± 6.1	0.64	0.02

**Table 4 sports-13-00357-t004:** Comparison of possession and passing statistics between champions Argentina (Copa America 2024) and Spain (Euro 2024).

Parameter	Argentina (Mean ± SD)	Spain (Mean ± SD)	*p*-Value (K.–S. Test)	Effect Size (Cohen’s *d*)
Possession per game (%)	55.7 ± 6.6	57.4 ± 8.1	0.70	0.23
Total passes per game	559.7 ± 99.7	619.3 ± 113.2	0.70	0.56
Pass length (m)	20.8 ± 13.4	19.8 ± 12.7	8.4 $\times \;10^{-4}$	0.08
Pass completion rate (%)	87.4 ± 3.3	89.5 ± 2.2	0.28	0.75

## Data Availability

The original contributions presented in this study are included in the article. Further inquiries can be directed to the corresponding author.

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
