# Peer review of "Comparative Analysis of Passing, Possession, and Goal-Scoring Trends in Euro 2024 and Copa America 2024"

_sports, 2025, doi:10.3390/sports13100357_

Round 1
Reviewer 1 Report
Comments and Suggestions for Authors
Thank you for the opportunity to review this manuscript. The authors have done a thorough job of examining the differences between the soccer styles of two of the premier tournaments. The results of this study could be transformative across levels within the practices of sports science, sports medicine, strength & conditioning, and coaching tactics.
Below are my comments and suggestions:
This study compares match strategies between Euro 2024 and Copa America 2024, analyzing passing, possession, and goal-scoring patterns using detailed event data from StatsBomb. Teams in Euro 2024 showed a more controlled style, with longer possession sequences, higher pass completion rates, and more frequent multi-pass combinations. In contrast, Copa America 2024 teams relied on longer individual passes and shorter possession phases, suggesting a more direct and fast-paced approach. While possession leading up to goals was greater in Euro matches, the distances from which goals were scored were similar across both tournaments. These results point to distinct regional approaches to gameplay that shape how matches unfold.
This manuscript presents a timely and well-structured comparative analysis of match strategies between Euro 2024 and Copa America 2024, focusing on passing, possession, and goal-scoring patterns. The hypothesis, that regional footballing philosophies influence tactical execution, is clearly testable and supported by the data. The methodology is sound, with appropriate statistical comparisons and clearly defined metrics.
- Relevance and Novelty: The topic is highly relevant to both sports science and performance analytics. To the best of my knowledge, this is the first study to directly compare tactical metrics between Euro 2024 and Copa America 2024 using high-resolution event data.
- Completeness: The manuscript covers key aspects of offensive play but could be more comprehensive by including defensive or transitional metrics.
- References: The citations are mostly recent and relevant. There is no evidence of excessive self-citation.
- Reproducibility: The methods are described in enough detail to allow replication, assuming access to the same dataset and software tools.
- Figures: Visualizations are clear and effectively support the findings. Statistical results are appropriately reported.
- Conclusions: The conclusions are consistent with the data and logically derived from the results.
- Ethics and Data Availability: Both statements are adequately addressed.
Recommendations and feedback
- Ensure that the references are consistently formatted
- Materials and Methods:
- Consider flow diagram of data processing
- Since the study relies heavily on custom MATLAB software to process Statsbomb data and compute possession, passing, and goal-scoring metrics, a mention of how the script was verified would strengthen the methodology section
- Please specify the version and company information of Statsbomb and MATLAB used for the custom analysis scripts.
- Statistical Analysis:
- Consider clarifying assumptions for parametric tests.
- Results:
- Consider using a Table to visualize statistical outcome(s).
- Consider decluttering the Figures 1 and 2.
- Discussion:
- Minimize repetitive themes.
- Pitch quality may or may not influence tactics.
- Consider adding context to the potential tournament-specific factors on tactics.
Author Response
We sincerely thank reviewer for the thoughtful and constructive feedback provided on our manuscript. The comments have been invaluable in helping us refine and strengthen the work. Below we provide a detailed, point-by-point response to each of the reviewer’s comments. In preparing the revised manuscript, we carefully addressed all points raised, with changes clearly indicated in the resubmitted files. Where appropriate, we have added new figures, tables, and clarifications to improve clarity, methodological rigor, and presentation.
Comment 1: Ensure that the references are consistently formatted
Response 1: We have revised the reference list to ensure format consistency.
Comment 2: Consider flow diagram of data processing.
Response 2: We have added a flow diagram (“new” Figure 1) that summarizes the data processing steps, from raw StatsBomb JSON match data through data import and harmonization, to analyses of possession timelines, passing features, and goal events, and finally to tournament-level outputs.
Comment 3: Since the study relies heavily on custom MATLAB software to process Statsbomb data and compute possession, passing, and goal-scoring metrics, a mention of how the script was verified would strengthen the methodology section
Response 3: We have clarified in the Methods that the outputs were cross-checked against independent sources. Specifically, for selected games and periods, possession timelines were manually compared with match video, confirming alignment of possession changes and durations. Goal timestamps were likewise verified against broadcast replays. We also ensured internal consistency (e.g., possession percentages summing to 100%) and cross-checked pass and goal counts with official match summaries. These steps give us confidence that the scripts accurately capture the underlying match events.
Comment 4: Please specify the version and company information of Statsbomb and MATLAB used for the custom analysis scripts
Response 4: We have added the versions of the MATLAB software and StatsBomb information.
Comment 5: Consider clarifying assumptions for parametric tests.
Response 5: We clarified in the Methods that parametric test assumptions were checked, and since distributions often deviated from normality, we emphasized nonparametric tests. For one comparison in which the values were time-dependent (pre-goal possession percentage), we used Welch’s t-test, which does not assume equal variances.
Comment 6: Consider using a Table to visualize statistical outcome(s).
Response 6: In line with this comment and those from Reviewer #2, we have added four tables to the manuscript that present direct comparisons of the statistical outcomes across the analyzed parameters.
Comment 7: Consider decluttering the Figures 1 and 2
Response 7: We performed some revisions in the figures and reduced the complexity without reducing the data reported in the figures.
Comment 8: Minimize repetitive themes (in Discussion).
Response 8: We have trimmed some sections of the discussion to avoid redundancy.
Comment 9: Pitch quality may or may not influence tactics.
Response 9: Thank you for this comment. We have mentioned pitch quality in the discussion.
Comment 10: Consider adding context to the potential tournament-specific factors on tactics.
Response 10: Thank you for this comment. We have mentioned the potential tournament-specific factors in the discussion.
Reviewer 2 Report
Comments and Suggestions for Authors
Dear authors,
Congratulations on your important research, which has the potential to make a significant contribution to soccer performance analysis. However, in order for your manuscript to be considered for publication, substantial revisions are required. You will find a detailed section-by-section evaluation of my comments and recommendations in the attached file.

Author Response
We sincerely thank the reviewer for the careful and constructive evaluation of our manuscript. The reviewer’s detailed comments have been extremely helpful in identifying areas for clarification, refinement, and restructuring. Below we provide a point-by-point response to each comment, highlighting the revisions made to the manuscript. All requested changes and corrections have been implemented, with modifications clearly marked in the resubmitted files. In particular, we have revised the abstract, streamlined keywords, removed irrelevant sections, added new tables, clarified methodological points, and improved consistency throughout the manuscript.
Comment 1: The Abstract is excessively long (over 400 words). A full revision is recommended, reducing the background and removing all descriptive statistics (mean, IQR, SD). Only the p-values should remain, and effect sizes should be added. The type of statistical tests performed should also be stated.
Response 1: We have revised the abstract in accordance with the reviewer’s recommendation. The new version is shorter, includes a reduced background, and reports only p-values and effect sizes, with descriptive statistics removed.
Comment 2: “Euro 2024” and “Copa America 2024” should be removed because they are already included in the title, as should “Public Health” because it is entirely unrelated to the article. What is “Public Health” doing in a paper that is clearly a technical-tactical analysis in soccer? Other keywords should be added to truly reflect the content of the paper, so that there are at least 5 keywords not appearing in the title.
Response 2: In the revised manuscript, the keywords “Euro 2024,” “Copa América 2024,” and “Public Health” have been removed. We have updated the keyword list to include terms that better reflect the scope of the study.
Comment 3: The second paragraph of the introduction, which concerns public health (lines 43–49), is entirely irrelevant to the article (and also has no references). I recommend removing it entirely
Response 3: We have removed this paragraph.
Comment 4: Likewise, the last paragraph after the stated aim is also completely irrelevant (and also without references). In the discussion, you could mention that different technical-tactical styles can also influence the physical demands on players, but even then it is a stretch to consider that this affects public health. This paragraph (lines 94–100) should be removed.
Response 4: We have removed these texts.
Comment 5: Support the statement “These stylistic differences are deeply rooted in the cultural and tactical histories of their respective regions, influencing match outcomes and broader tournament trends” with references.
Response 5: We have included the study by Izquierdo and Redondo (2022) to support this quote.
Comment 6: In the paragraph stating the aim of the study (introduction), clearly specify the research questions investigated
Response 6: Done.
Comment 7: The significance level (p<0.001) is too strict for sports studies and is not justified by the authors.
Response: In our dataset, all p-values fall either below 0.001 or well above 0.05. Thus, the choice of significance threshold (whether 0.05, 0.01, or 0.001) does not alter the conclusions of the study. Now, we reported results at p<0.05 for consistency with other sports studies. To clarify this point, we have revised the Statistical Methods section to note that the interpretation of results remains unchanged under all conventional thresholds, and we also provide effect sizes alongside p-values to reflect the magnitude and practical relevance of the differences.
Comment 8: There is no clear mention of effect sizes, which are important for interpreting practical significance.
Response 8: In the revised manuscript, we have added a description of Cohen’s d in the Materials and Methods section. All new tables now report effect sizes alongside p-values, and the comparisons are interpreted on the basis of both statistical significance and practical significance (based on Cohen’s d).
Comment 9: There is no mention of how matches with extra time were handled.
Response 9: In the revised manuscript, we have clarified that extra time was included in all analyses of possession, passing, and goal-scoring statistics. Extra time represented only a small fraction of the data, as it occurred in just one Copa America 2024 match (the final match) and only in knockout-stage matches in Euro 2024. Consequently, its contribution to the overall statistics was small. We note that one figure examining pass completion rate as a function of time includes only the first and second halves, as indicated in the caption, and does not incorporate extra time.
Comment 10: The last paragraph (lines 222–225) is unrelated to the sub-section (Statistical Analysis Approach) in which it is located.
Response 10: In the revised manuscript, we have created a new subsection titled Ethical Considerations and moved this paragraph there.
Comment 11: Descriptive statistics are completely absent from the study. It is paradoxical that in an article with so many numbers and statistical analyses there is not a single table! It is essential to create: A table for continuous variables, including means and SDs separately for Euro and Copa America, the test performed, statistical significance, and effect size. A table for categorical variables, including percentages for each category separately for Euro and Copa America, the test performed, statistical significance, and effect size. Corresponding tables should also be created for comparisons between Spain and Argentina.
Response 11: Thank you for this important comment. The expert reviewer is right. In the revised manuscript, we have added tables for the data that are best presented in tabular format. These tables include continuous and categorical variables, with descriptive statistics (means, SDs, and percentages), the statistical tests performed, and both statistical significance and effect sizes for all comparisons. Corresponding tables for Spain and Argentina have also been added in line with the reviewer’s recommendation.
Comment 12: Greater connection of the findings with the theoretical framework (game models, playing styles).
Response 12: We have improved the connection of the findings with the theorical framework of the study.
Comment 13: Use of references for the following sentences in the “Limitations” sub-section:
- “National team tournaments occur under unique conditions, with limited preparation time and squad rotation, which may not reflect the day-to-day tactical behaviors observed in domestic leagues.”:
Response 13a: We added a reference to support this statement. - “Third, the growing number of American players competing in European leagues may lead to a degree of stylistic convergence, as these players are increasingly influenced by European tactical and training methodologies, potentially diluting regional distinctions.”
Response 13b: We added a reference to support this statement. - “Finally, while the study identified differences in possession, passing, and goal-scoring trends, it relied on aggregate tournament data and did not include qualitative tactical analysis, which could offer a deeper understanding of decision-making and in-game adaptations.”
Response 13c: This is a specific limitation of our study in terms of the scope of our investigations, we believe there is no need to cite a publicly available reference for this statement.
Comment 14: Removal of the sub-section “4.5. Public Health Implications of Tactical Differences” because it is off-topic; the conclusions are too indirect (if there were direct physical performance variables in the paper, it might be justified). A brief mention could be made either in the Discussion or Conclusions, but certainly not as a standalone sub-section for such an indirect effect that technical-tactical differences might have on public health. Furthermore, this brief mention should refer to the health of the athletes (in this case, football players) and not public health.
Response 14: We have removed this section.
Comment 15: The Conclusions section is adequate for a basic summary, but again caution is needed with states to public health.
Response 15: We have removed the section about public health.
Round 2
Reviewer 2 Report
Comments and Suggestions for Authors
The authors have addressed all comments, resulting in a significant improvement in the quality of the manuscript. I have only one minor observation to make regarding the fact that the Materials and Methods section does not mention Cohen's d thresholds for small, medium and large effect sizes.
Author Response
Comment 1: The authors have addressed all comments, resulting in a significant improvement in the quality of the manuscript.
Response 1: Thank you for your thoughtful suggestions, which have improved the quality of our manuscript.
Comment 2: I have only one minor observation to make regarding the fact that the Materials and Methods section does not mention Cohen's d thresholds for small, medium and large effect sizes.
Response 2: Thank you for noting this oversight. We have revised the text to include the interpretation of Cohen’s d and the corresponding thresholds.